# Application of Deep Learning in Predicting Particle Concentration of Gas–Solid Two-Phase Flow

**Zhiyong Wang** [1,2]**, Bing Yan** [1,2,3,*] **and Haoquan Wang** [1]

1   School of Information and Communication Engineering, North University of China, Taiyuan 030051, China; sz202205016@st.nuc.edu.cn (Z.W.); wanghaoquan12@163.com (H.W.)
2   China-UK Joint Laboratory of Particle & Two-Phase Flow Measurement Technology, North University of China, Taiyuan 030051, China
3   Shanxi Key Laboratory of Signal Capturing & Processing, North University of China, Taiyuan 030051, China
*   Correspondence: yanbing122530@126.com

**Abstract:** Particle concentration is an important parameter for describing the state of gas–solid two-phase flow. This study compares the performance of three methods, namely, Back-Propagation Neural Networks (BPNNs), Recurrent Neural Networks (RNNs), and Long Short-Term Memory (LSTM), in handling gas–solid two-phase flow data. The experiment utilized seven parameters, including temperature, humidity, upstream and downstream sensor signals, delay, pressure difference, and particle concentration, as the dataset. The evaluation metrics, such as prediction accuracy, were used for comparative analysis by the experimenters. The experiment results indicate that the prediction accuracies of the RNN, LSTM, and BPNN experiments were 92.4%, 92.7%, and 92.5%, respectively. Future research can focus on further optimizing the performance of the BPNN, RNN, and LSTM to enhance the accuracy and efficiency of gas–solid two-phase flow data processing.

**Keywords:** gas–solid two-phase flow data processing; BPNN; RNN; LSTM

## 1. Introduction

Gas–solid two-phase flow is a common and complex fluid state in industries such as energy and chemical engineering [1]. Particle concentration is a key parameter that determines the flow characteristics of gas–solid two-phase flow and plays a crucial role in investigating these characteristics and optimizing industrial production processes. Various techniques, including microwave [2], capacitance [3], acoustic [4], and optical wave fluctuation [5] techniques, have been proposed for measuring the parameters of gas–solid two-phase flow. Particularly, the electrostatic principle has received widespread attention in recent years due to its reliability and high sensitivity. Nonintrusive sensors are widely used for detecting charge in various industrial applications [6,7].

Traditional methods extract useful signals from electrostatic signals, using different algorithms to accurately study the parameters of gas–solid two-phase flow. For instance, Wang et al. decomposed the signal from an electrostatic sensor using harmonic wavelet transform (HWT) [8] and discrete wavelet transform (DWT) [9]. Zhang et al. utilized the Hilbert–Huang transform to obtain the average flow characteristic parameters of particles within the sensor, such as average flow velocity and average mass flow rate [10]. However, a challenge of electrostatic sensing technology is establishing a model between particle concentration, flow rate, and electrostatic current signal. This is due to the complexity of the electrostatic behavior of powder particles, as well as the amount and polarity of charges being related not only to the properties of the particles themselves (shape, size, distribution, roughness, relative humidity, chemical composition, etc.) but also to the material and arrangement of the pipeline, as well as conveying the conditions of particles within the pipeline (pipe size, conveying velocity) [11,12]. Researchers have improved the characteristics of the instrument to mitigate the influence of sensors on flow patterns



and enhance the consistency of spatial sensitivity, thereby improving measurement accuracy [9,10]. This compensation for spatial sensitivity is an important step in enhancing measurement accuracy and improving the instrument's suitability for flow patterns.

From a theoretical perspective, it is still challenging to explain the complexity and randomness of gas–solid and other multiphase flow systems. Therefore, it is crucial to acquire a large amount of data through experiments and in actual production processes, study the phenomena using statistical methods, and establish models. Modeling unknown functions of unrelated variables using machine learning techniques is an effective application of electrostatic sensor modeling. Deep learning algorithms can effectively model variables such as signals, concentration, and particle velocity. Furthermore, deep learning [13–15] offers valuable characteristics that allow for efficient learning and processing of complex relationships and non-linear features in data, providing efficient and accurate modeling and prediction capabilities.

In the field of measurement research on gas–solid two-phase flow, Yan et al. have made effective explorations in applying machine learning algorithms to optimize models and improve measurement accuracy [16]. Despite this progress, there has been relatively little research on using deep learning methods to determine parameters in gas–solid two-phase flow.

Nevertheless, deep learning models demonstrate sufficient flexibility to promptly respond to and update based on changes in data in order to adapt to dynamic system variations. This characteristic provides robust support for industrial production optimization, environmental protection, and process safety. Despite the limited research on deep learning for determining parameters in gas–solid two-phase flow, it is foreseeable that deep learning will emerge as a pivotal method in future studies, offering new perspectives and opportunities for addressing related issues.

## 2. Materials and Methods

Deep learning is a sub-field of artificial intelligence that focuses on developing algorithms and models capable of learning and making predictions or decisions without explicit programming. It involves studying statistical models and algorithms that enable computers to automatically analyze and interpret complex patterns and relationships in data.

At the core of machine learning is the construction of computational models that can learn from data and make predictions or decisions based on that knowledge. These models are trained using large datasets, which consist of input data and corresponding desired outputs or outcomes. During the training process, the models learn to recognize patterns, extract meaningful features, and generalize from the data to make predictions or decisions on new, unseen data.

In the context of complex gas–solid two-phase flow data, this study utilized several models for predicting particle concentration, including RNNs, LSTM, and BPNNs.

### 2.1. BPNN

The main steps of training a BPNN include parameter initialization, forward propagation, loss computation, back propagation, and parameter updating [17–19]. During forward propagation, input samples are processed through the network to obtain output results. The loss function is then calculated to measure the discrepancy between the output and the target. Subsequently, back propagation is performed to compute the gradients layer by layer and update the weights and biases. This iterative process continuously adjusts the parameters to make the network output approach the target values.

### 2.1.1. Forward Propagation

During forward propagation, input data are transmitted from the input layer to the output layer. At each layer, the input is multiplied by weights and passed through

an activation function to produce the output. The process can be described using the following equations:

$$Z_i = \sum_{i=1}^{n} (W_i A_{i-1} + b_i) \tag{1}$$

$$y_i = f(Z_i) \tag{2}$$

In these equations, $Z_i$ represents the weighted sum of inputs at layer $i$, $W_i$ represents the weighted matrix connecting layer $i-1$ to layer $i$, $A_{i-1}$ represents the output of layer $i-1$, $b_i$ represents the bias vector at layer $i$, $f()$ represents the activation function, and $A_i$ represents the output of layer $i$.

### 2.1.2. Backward Propagation

Backward propagation is used to calculate the gradients of the parameters (weights and biases) with respect to the loss function. This allows the neural network to update its parameters and improve its performance. The gradients are calculated using the chain rule of differentiation.

For example, let us consider the output layer. Assuming the activation function is $f()$, the loss function is $L$, the input to the output layer is $Z$, and the output is A. The gradient of the loss function with respect to the output can be calculated as:

$$\frac{\partial L}{\partial Z} = \frac{\partial L}{\partial a} f'(Z) \tag{3}$$

Here, $f'$ represents the derivative of the activation function. Using these gradients, the weights and biases can be updated according to the following formulas:

$$w_l = w_l - a\frac{\partial L}{\partial w_l} \tag{4}$$

$$b_l = b_l - a\frac{\partial L}{\partial b_l} \tag{5}$$

In these formulas, $a$ represents the learning rate of the neural network, $\partial L/\partial w_l$ represents the gradient of the loss function with respect to the weights, and $\partial L/\partial b_l$ represents the gradient of the loss function with respect to the biases.

The common activation functions and their expressions are as follows:

$$f_{Sigmoid} = \frac{1}{1 + e^{-x}} \tag{6}$$

The Sigmoid function can map real numbers to between 0 and 1 in the input and is usually used for binary classification problems.

$$f_{ReLU} = Max(0, x) \tag{7}$$

The Re*LU* function returns the input itself for non-negative inputs and returns 0 for negative inputs. Re*LU* is widely used in deep learning because it can accelerate training and reduce the risk of overfitting.

$$f_{Tanh} = \frac{e^x - e^{-x}}{e^x + e^{-x}} \tag{8}$$

The Tanh function is similar to the Sigmoid function but maps the input to between $-1$ and 1 and is usually used for multi-classification problems.

These activation functions introduce non-linearity into the neural network, allowing it to learn complex patterns and make predictions. The choice of activation function depends on the nature of the problem and the characteristics of the data.

### 2.1.3. Advantages and Disadvantages of BPNNs

BPNNs have advantages in handling non-linear relationships, flexibility, and adaptability. By optimizing model parameters, it effectively reduces the loss function and improves performance. However, BPNNs have the drawbacks of long training time, susceptibility to local minimums, and sensitivity to outliers, requiring data preprocessing and parameter tuning to avoid overfitting. When dealing with gas–solid two-phase flow data, additional parameter tuning and preprocessing may be necessary to enhance stability and accuracy.

### 2.2. RNNs and LSTMs

RNNs and LSTMs are neural network architectures used for processing sequential data [20]. RNNs have recurrent connections that allow information to be passed and shared within a sequence, capturing temporal dependencies and contextual information. However, traditional RNNs suffer from the issues of vanishing and exploding gradients. To address this problem, LSTM was introduced, which incorporates gate mechanisms to selectively update, retain, or discard information, mitigating the gradient problem and better capturing long-term dependencies. Hence, LSTM can be regarded as an enhanced version of RNN designed to improve the handling of long sequential data.

### 2.2.1. RNNs

An RNN (Recurrent Neural Network) is a type of recursive neural network used for processing sequential data. Its principle can be represented by the following equation:

$$h_t = f(W_{hh} + W_{xh}x_t + b_h) \tag{9}$$

Here, $h_t$ represents the hidden state at time step $t$, $x_t$ represents an element of the input sequence, $W_{hh}$ is the weight matrix from hidden state to hidden state, $W_{xh}$ is the weight matrix from input to hidden state, $b_h$ is the bias vector, and $f$ is the activation function.

### 2.2.2. LSTM

LSTM (Long Short-Term Memory) is a variant of Recurrent Neural Networks (RNNs) that effectively handles long-term dependencies. It achieves this by incorporating gate mechanisms to control the flow of information, which primarily consists of input gate, forget gate, and output gate [21,22], as follows:

1.    Forget gate.

The forget gate determines which information from the previous time step's memory state should be forgotten. It is calculated using the following formula:

$$g_f = \delta(w_f[h_{t-1}, x_t] + b_f) \tag{10}$$

Here, $w_f$ and $b_f$ are the parameters of the forget gate, and $\delta$ represents the Sigmoid activation function; $h_{t-1}$ refers to the previous time step's hidden state, and $x_t$ represents the current input.

2.    Input gate.

The input gate determines which information from the current time step should be updated into the memory state. It is calculated using the following formula:

$$i_t = \delta(w_i[h_{t-1}, x_t] + b_i) \tag{11}$$

$$g_t = \tanh(w_g[h_{t-1}, x_t] + b_g) \tag{12}$$

Here, $i_t$ denotes the output of the input gate, $g_t$ represents the candidate memory value at the current time step, and $w_i$, $w_g$, $b_i$, and $b_g$ are the weights and biases of the input gate.

3. Updating memory state (cell state).

The previous memory state, $c_{t-1}$, is updated based on the outputs of the forget gate and input gate. The computation formula is as follows:

$$c_t = \delta(w_f[h_{t-1}, x_t] + b_f) \tag{13}$$

Here, $c_t$ represents the current memory state.

4. Output gate.

The output gate determines which information from the current hidden state should be output to the next time step or externally. It is calculated using the following formula:

$$o_t = \delta(w_o[h_{t-1}, x_t] + b_o) \tag{14}$$

$$h_t = o_t\tanh(c_t) \tag{15}$$

Here, $O_t$ denotes the output of the output gate, and $h_t$ represents the current hidden state.

At each time step, LSTM utilizes the input, previous hidden state, and memory state to update the memory state and hidden state through the computation of the forget gate, input gate, and output gate. This enables LSTM to model and retain information from sequential data.

### 2.2.3. Advantages and Disadvantages of RNNs and LSTM

RNNs can effectively capture temporal dependencies in the time-series data of gas–solid two-phase flow, enabling improved prediction and analysis. Its strength lies in its ability to capture sequential time-dependent information, making it suitable for handling time-series data in gas–solid two-phase flow. However, it suffers from the challenge of vanishing or exploding gradients when dealing with long sequences. Consequently, its performance may be limited when handling extremely long sequences.

On the other hand, LSTM's memory units allow for selective retention and forgetting of information, making it a suitable choice for addressing long-term dependencies in gas–solid two-phase flow data. It overcomes the issue of long-term dependencies in RNNs and handles temporal data more effectively in gas–solid two-phase flow. Compared to traditional RNNs, it performs better in handling long sequences and long-term dependencies. Nevertheless, it may require increased computational resources and training time.

## 3. Data Collection and Processing

### 3.1. Data Collection

The experiment required the collection of data such as temperature, humidity, pressure difference, and velocity during the running process of the particles. The experimental equipment utilized a gas–solid two-phase flow detection device provided by the laboratory to complete the research work. The experimental platform is shown in Figure 1.

The experimental platform equipment included a separator, a receiving bin, a feeding bin, a blower, a power supply unit, and a digital multimeter, as well as a ring-shaped electrostatic sensor, temperature and humidity sensors, and a pressure difference sensor located near the annular electrostatic sensor. The experiment used fly ash particles with particle sizes ranging from 0.1 mm to 0.9 mm.

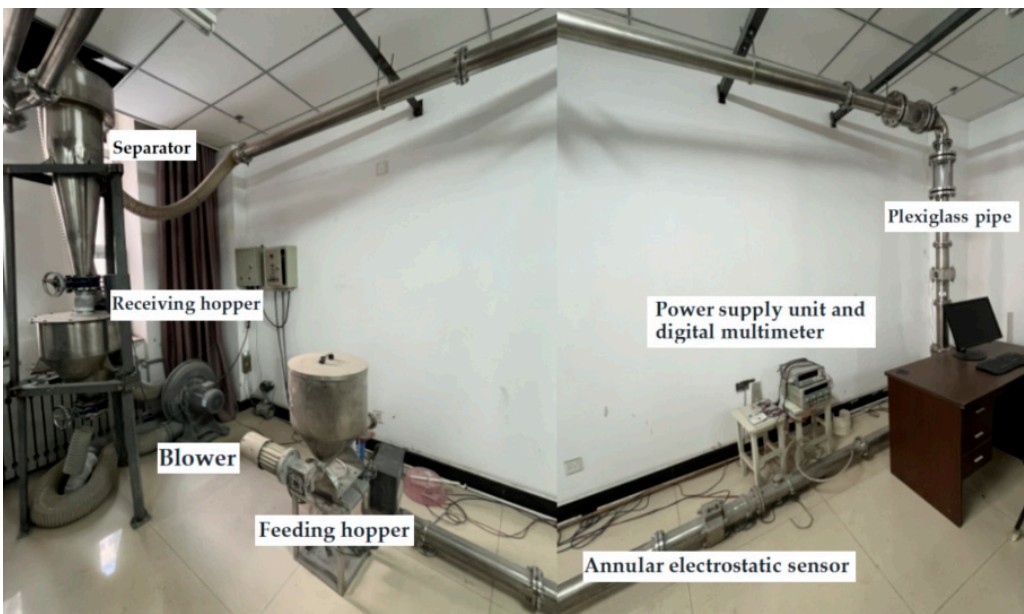

**Figure 1.** Experimental platform for gas–solid two-phase flow.

The following equipment was used for measurement and monitoring in this experimental platform:

1.    KZWSRS485 temperature and humidity transmitter.

The Kunlun Zhongda Company in Beijing, China manufactures this transmitter, which comprises a sensor, signal-processing circuit, and communication interface. It converts the measured temperature and humidity data into standard electrical signal output. The temperature measurement range is from 0 °C to 50 °C, and the humidity measurement range is from 0%RH to 100%RH. The KZWSRS485 temperature and humidity transmitter have high accuracy, reliability, and stability and can operate within a wide range of temperatures and  humidity.

2.    KZY-808BGA pressure difference sensor.

The sensor, produced by Kunlun Zhongda Company in Beijing, China, is designed for measuring air pressure differences and converting them into corresponding electrical signal outputs. It offers a measurement range from 0 to 3 KPa.

3.    Electrostatic sensor.

Two inductive ring-shaped electrostatic sensors and a metal shield were used in the experiment. The sensor electrodes were made of highly sensitive stainless steel material in a ring-shaped structure, providing good wear resistance. To ensure the stability of the signal output, the electrostatic sensor was equipped with a metal shield to reduce the influence of external electromagnetic interference.

4.    APS3005S-3D power supply unit.

This device is manufactured by Shenzhen AntaiXin Technology Co., Ltd., located in Shenzhen, China. It provides stable voltage and current outputs for sensors, ensuring the normal operation of the sensors.This device provided stable voltage and current output to the sensors, ensuring their normal operation.

5.    GDM-842 digital multimeter.

The manufacturer of the GDM-842 Digital Multimeter is GW Instek (Good Will Instrument Co., Ltd.), located in Taiwan, China. This multimeter is used to measure voltage signals outputted by sensors, ensuring accurate measurement results.

The use of these devices in the experimental platform aimed at achieving precise measurements and data acquisition of environmental conditions, providing reliable data support for the experiment.

Due to the randomness and complexity of fluid motion, the charge signals sensed by electrostatic sensors often exhibit instability. Processing the charge signal itself is relatively challenging. Therefore, the experiments were designed to consider the influence of the sensor's impedance and signal bandwidth, and corresponding conditioning circuits were developed. Through the conditioning circuit, the charge signal was able to be converted into a stable and measurable voltage signal.

The data acquisition system uses an FPGA (Intel's EP4CE40F) as the main control chip, which is produced by Shenzhen Gongshen Electronic Technology Co., Ltd. located in Shenzhen, China. It employs the AD7606 as the ADC with a precision of 16 bits and a sampling frequency set at $10^4$ Hz. In addition, to achieve data storage, a NAND flash memory was added, utilizing the H27U1G8F2B chip from Micron Technology Inc., located in Boise, ID, USA. The CY60813A chip from Cypress Semiconductor Corporation in San Jose, CA, USA, was utilized as the core chip for USB transmission. This chip played a crucial role in enabling seamless communication with the PC. The data collection process is illustrated in Figure 2.

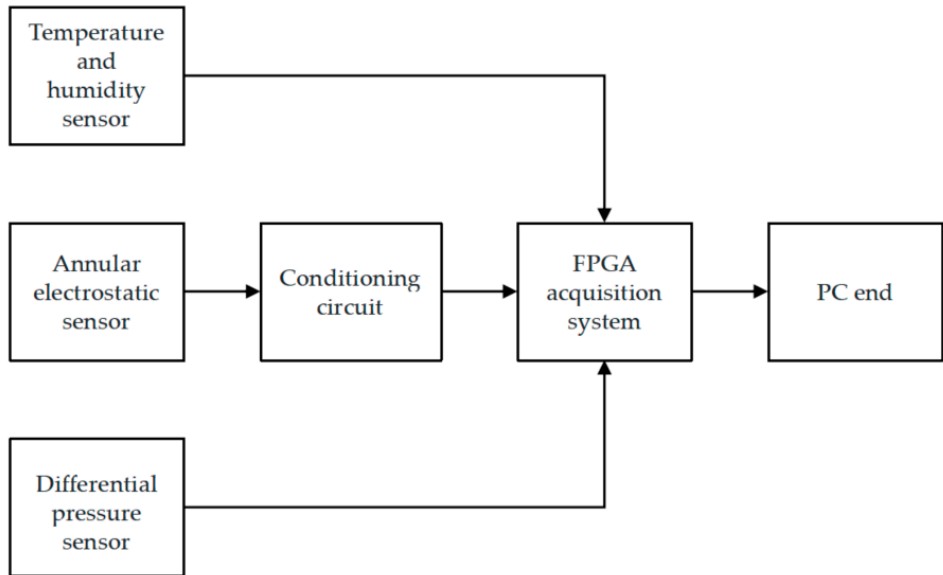

**Figure 2.** Data collection process diagram.

The experiment utilized a feeder and a blower to control the mass flow rate and velocity of particles, obtaining data on gas–solid two-phase flow at different mass flow rates and velocities within a range of 30–50 m/s for velocity and 33.5–54.5 g/s for mass flow rate.

### 3.2. Calculation of Particle Velocity and Particle Concentration

The experiment involved the processing of voltage output from two electrostatic sensors. After running the system for a period of time, it was observed that the signal outputs between these two sensors exhibited notable similarities. This trend can be clearly observed in Figure 3.

The experiment measured the difference in the index numbers corresponding to the peak values of signals from the upstream and downstream sensors as the particle delay. Considering the known constant values of the sampling frequency and the distance between the two electrostatic sensors, this delay value can be used as an approximate particle velocity.

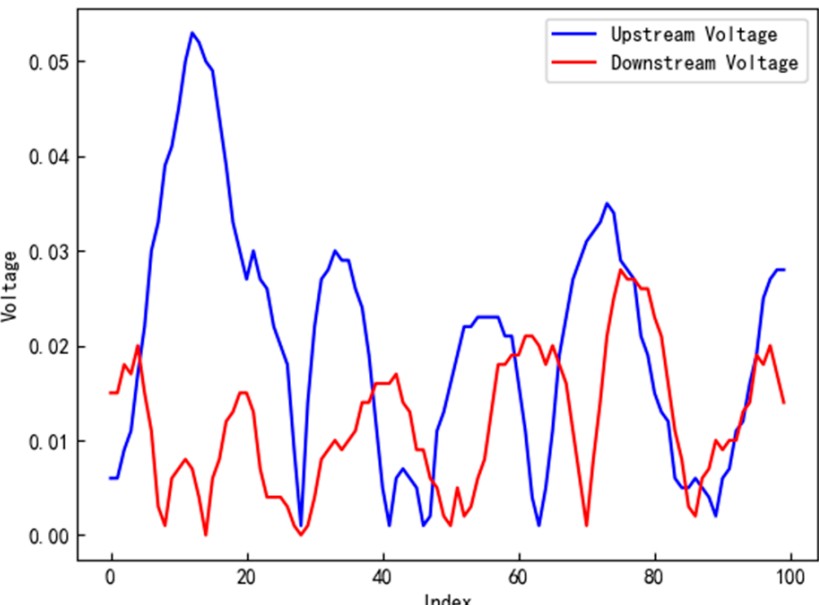

**Figure 3.** Comparison of signal outputs between upstream and downstream sensors.

After obtaining the velocity, the particle concentration can be calculated using the following formula:

$$C = \frac{q}{AV} \tag{16}$$

where $C$ is the particle concentration, $q$ is the mass flow rate, $V$ is the particle velocity, and $A$ is the cross-sectional area of the particle transport pipeline.

By following the aforementioned process, the experiment obtained complete data for temperature, pressure difference, humidity, signals from upstream and downstream sensors, velocity, and particle concentration.

### 3.3. Normalization of Model Data

Data normalization is typically performed in deep learning models to ensure that the input data have similar value ranges across different features. This helps in accelerating model convergence and improving stability. Normalization prevents issues such as feature bias and excessive gradient changes, allowing the model to treat all features fairly and enhance training effectiveness and reliability. It is calculated using the following formula:

$$x' = \frac{x - x_{\min}}{x_{\max} - x_{\min}} \tag{17}$$

where, $x_{\min}$ refers to the minimum value of the original data, and $x_{\max}$ refers to the maximum value of the original data. $x$ refers to the value before normalization, and $x'$ refers to the value after normalization.

### 4. Results

#### 4.1. Experimental Preparation and Environment Configuration

The experiment utilized a two-phase gas–solid flow dataset consisting of seven columns of data, including temperature, humidity, upstream sensor signal, downstream sensor signal, delay, pressure differential, and particle concentration. The training set consisted of 25,500 samples with varying velocities and particle concentrations. The test set was extracted from the remaining samples and contained 5100 samples. The training set was used for model training and parameter optimization, while the test set was used to evaluate the model's predictive performance.

To establish the experimental environment, Jupyter Lab (4.0.3) was chosen as the computational environment, which is a scientific computing tool implemented in an interactive manner. For the selection of the main programming language, Python (3.9) was adopted, which is a high-level programming language widely used in the field of machine learning.

To enhance the reliability of project management and the independence of experiments, the Anaconda platform was utilized. With Anaconda, it is possible to create isolated virtual environments to ensure the isolation of dependencies between different projects. Within the virtual environment, TensorFlow was installed as the machine learning library, which is an open-source framework developed by Google. Along with TensorFlow, other essential machine learning libraries and tools such as Keras, PyTorch, and scikit-learn experiments were installed to enhance the functionality and flexibility of the experiments.

*4.2. Model Evaluation Metrics*

This experiment set up three evaluation metrics: prediction accuracy ($A_F$), mean squared error ($E_{MSE}$), root-mean-square error ($E_{RMSE}$), and mean absolute error ($E_{MAE}$). The specific expressions are:

$$A_F = \frac{1}{n}\sum 1 - \frac{|y - y'|}{y} \tag{18}$$

$$E_{RMSE} = \sqrt{\frac{1}{n}\sum (y - y')^2} \tag{19}$$

$$E_{MSE} = \frac{1}{n}\sum (y - y')^2 \tag{20}$$

$$E_{MAE} = \frac{1}{n}\sum |y - y'| \tag{21}$$

where $n$ represents the total number of samples, and $y$ and $y'$ represent the actual value and predicted value, respectively.

*4.3. Model Construction and Parameter Determination*

In order to construct a model for predicting particle concentration, the present experiment used temperature, humidity, upstream and downstream sensor signals, velocity, and pressure difference as input parameters and the predicted particle concentration as the output of the model. The experiment employed BPNN, RNN, and LSTM models for modeling. Before beginning the modeling process, it was necessary to determine the number of hidden layers and the number of nodes in the hidden layers.

The decision to utilize three hidden layers in the experimental setup was driven by the findings of preliminary experiments, which suggested that a model with three hidden layers exhibits superior capability in capturing the intricate features within the dataset. Conversely, the potential drawbacks associated with introducing a fourth hidden layer, such as an increased risk of overfitting, heightened model complexity without substantial performance enhancements, and escalated computational expenses, were taken into consideration. The primary focus of the experiment was to explore the impact of varying node quantities within the three hidden layers. Starting with 40 nodes, the quantity was incrementally increased to 80 in increments of 10 nodes. Evaluation of each parameter combination was conducted to pinpoint the combination of node quantities that yielded the most favorable results. Throughout the experiment, $E_{MSE}$ was employed as the designated loss function.

Following the completion of training, the model's performance was evaluated using the test set. The prediction accuracy of models with different parameter settings was quantified by calculating the loss function. The experimental results are shown in Figures 4–6.

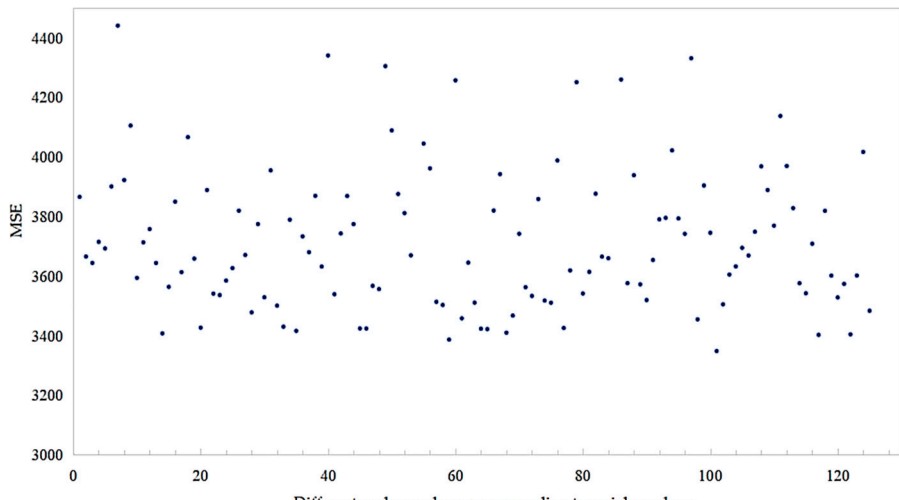

**Figure 4.** $E_{MSE}$ comparison of different LSTM network structures.

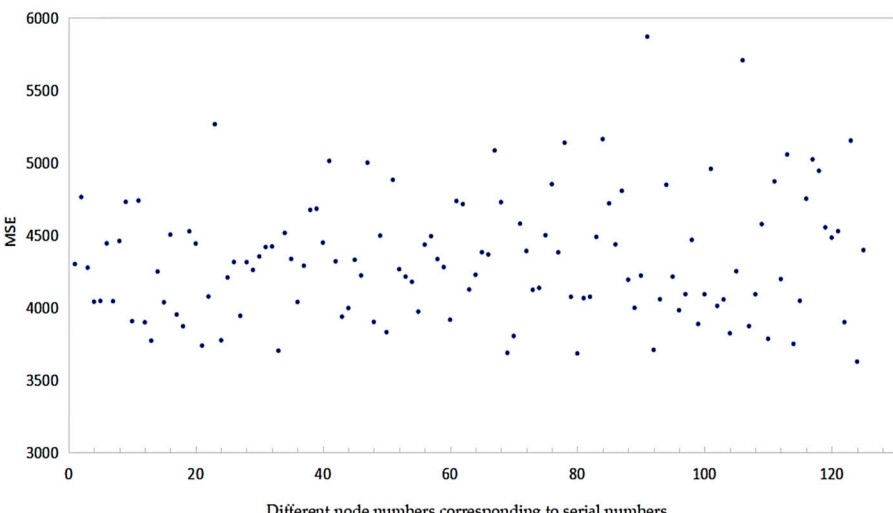

**Figure 5.** $E_{MSE}$ comparison of different RNN structures.

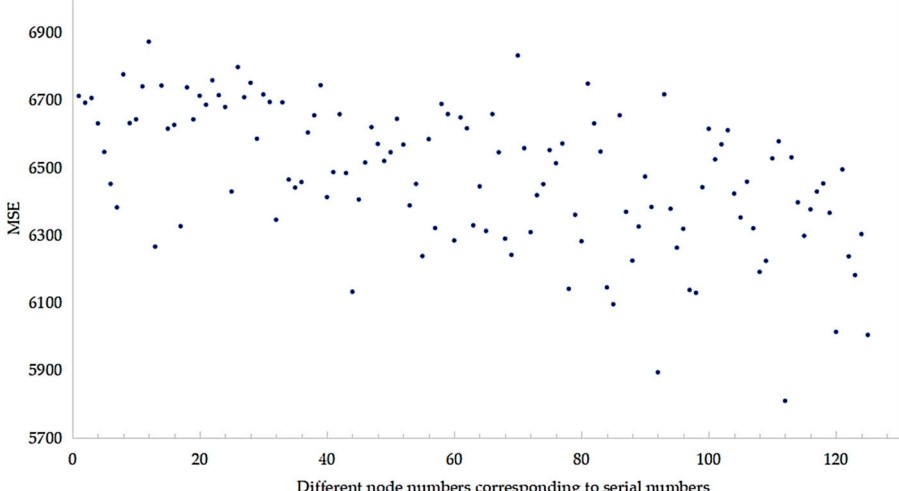

**Figure 6.** $E_{MSE}$ comparison of different BPNN structures.

Based on the results presented in Figures 4–6, it is evident that the BPNN demonstrated superior predictive performance when configured with three hidden layers and 80, 60, and 50 memory units in each respective layer. Similarly, the LSTM model delivered optimal results with three hidden layers and 80, 80, and 70 memory units, while the RNN model exhibited the best predictive performance when utilizing three hidden layers with 80, 40, and 40 memory units. The decision to set the maximum iteration count to 500 was informed by experimental data indicating that errors across different models converged to a stable level after 500 iterations. The parameters determined for the BPNN, LSTM, and RNN models are presented in Table 1.

**Table 1.** The key parameters of the model.

| Parameter | BPNN | RNN | LSTM |
|---|---|---|---|
| Number of Hidden Layers | 3 | 3 | 3 |
| Number of Units per Layer | 80, 60, 50 | 80, 40, 40 | 80, 80, 70 |
| Maximum Number of Iterations | 500 | 500 | 500 |
| Training Batch Size | 32 | 32 | 32 |

*4.4. Results*

When handling gas–solid two-phase flow data, the present experiment compared the predictive performance of different methods and used three evaluation metrics for comparison, including $A_F$, $E_{RMSE}$, and $E_{MAE}$. Table 2 shows the prediction results of the different models. Figure 7 presents a comparison between the predicted values and actual values for the different models.

**Table 2.** Comparison of prediction results for different models.

| Prediction Model | $A_F$ | $R_{MSE}$ | $R_{MAE}$ |
|---|---|---|---|
| BPNN | 92.5 | 52.22 | 37.45 |
| RNN | 92.4 | 53.19 | 38.88 |
| LSTM | 92.7 | 53.43 | 38.70 |

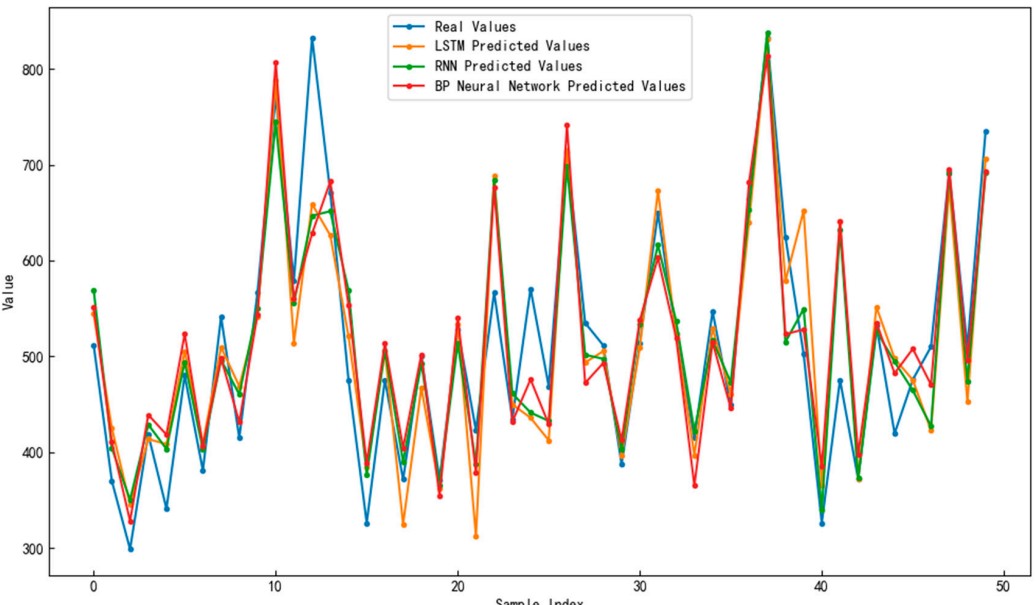

**Figure 7.** Comparison of predicted values and actual values for different models.

After obtaining the predicted values and actual values of the different models' test sets, the results were divided into different intervals based on the relative error. By evaluating

the proportion of low-error-interval samples, the performance of the models was able to be quantitatively assessed. A high proportion of low-error-interval samples indicated that the model could accurately predict the majority of samples, demonstrating a high level of precision. The proportions of samples within different relative error intervals for each model can be observed in Figure 8.

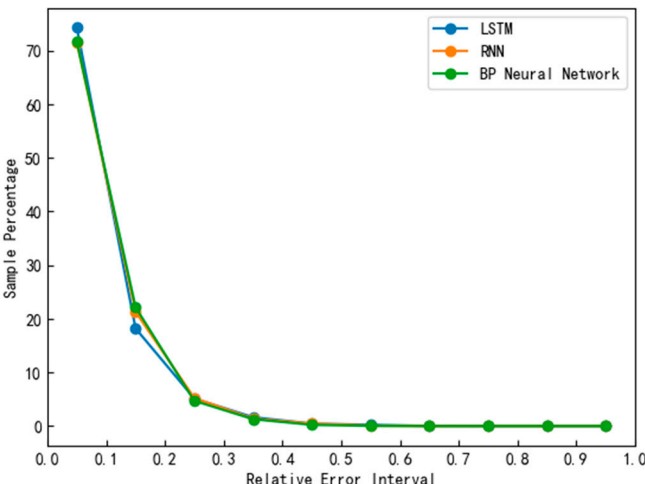

**Figure 8.** Relative error distribution plots of different models.

Based on the analysis of experimental results, it was evident that there were differences in the performance of the LSTM, RNN, and BPNN models in predicting the relative error of particulate matter concentration in gas–solid two-phase flow. In the relative error range of [0.0, 0.1), the LSTM model demonstrated strong performance, with an accuracy of 74.27%, which was slightly higher than the RNN model (71.57%) and the BPNN model (71.61%). Regarding the overall prediction performance indicators, the three models showed similar values in prediction accuracy, root-mean-square error, and mean absolute error, with no significant advantages observed. The LSTM model exhibited superior memory and long-term dependency capabilities compared to the BPNN and RNN, making it suitable for handling time-series data and long-distance dependency relationships, thereby providing the LSTM model with stronger modeling capabilities for gas–solid two-phase flow data processing. Although the LSTM model required a longer training time and computational resources, its training efficiency and convergence speed in handling complex gas–solid two-phase flow sequence data were higher compared to the BPNN and RNN, as observed in Figures 3–5, where, under the same number of iterations (100) and parameter settings, the LSTM's loss function was relatively smaller than those of the BPNN and RNN.

In summary, according to the experimental results, the LSTM model performs well in the relative error range of [0.0, 0.1), which is possibly attributable to its superior memory and long-term dependency capabilities, as well as its modeling capabilities in handling gas–solid two-phase flow sequence data. However, there were no significant differences observed among the three models in terms of overall prediction performance indicators.

## 5. Discussion

This experimental study compared the performance of LSTM, RNN, and BPNN models in predicting the concentration of particulate matter in gas–solid two-phase flow. The results showed that within the relative error range of [0.0, 0.1), the LSTM model exhibited the best performance, validating its advantages in handling time-series data and capturing long-term dependencies. Although the overall prediction performance indicators of the three models were similar and showed no significant advantages, they all demonstrated certain predictive capabilities and adaptability. This may be due to the similar challenges and limitations they face in handling the concentration of particulate matter in gas–solid two-phase flow. The complexity and dynamics of gas–solid two-phase flow may lead to

noise and uncertainty in the data, posing similar challenges for all models. Additionally, the prediction of particulate matter concentration is influenced by various factors, such as gas flow velocity, particle size distribution, and pipeline or equipment structure, which may have similar impacts on the prediction performance of different models.

Furthermore, these models possess similar modeling and expressive capabilities, leading to similar predictive performance when processing gas–solid two-phase flow data. The quantity and quality of the data may have similar effects on the prediction performance of the three models, resulting in relatively close predictive effects if the dataset features are challenging to all of the models to some extent. Therefore, despite potential differences in handling time-series data and capturing long-term dependencies, these models demonstrate similar predictive effects in specific tasks of predicting the concentration of particulate matter in gas–solid two-phase flow.

To enhance the performance of these models, it is recommended to conduct further experiments with different parameter settings and optimization strategies, feature selections and engineering, and model integration methods. Future research may involve evaluating other machine learning models (such as CNNs and self-attention mechanism models), improving feature selection and engineering methods, and exploring the interpretability of the models. Interdisciplinary cooperation is crucial to integrate fluid mechanics and deep learning for in-depth research to seek more reliable and accurate data processing models for gas–solid two-phase flow.

**Author Contributions:** Conceptualization, H.W.; Methodology, Z.W.; Software, Z.W.; Resources, B.Y.; Data curation, B.Y.; Writing—original draft, Z.W.; Writing—review & editing, B.Y.; Supervision, H.W. All authors have read and agreed to the published version of the manuscript.

**Funding:** This work was supported by the Open Research Fund of the Shanxi Key Laboratory of Signal Capturing and Processing.

**Data Availability Statement:** Data are contained within the article.

**Conflicts of Interest:** The authors declare no conflict of interest.

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
