# Peer review of "Application of Deep Learning in Predicting Particle Concentration of Gas–Solid Two-Phase Flow"

_fluids, doi:10.3390/fluids9030059_

Round 1

Reviewer 1 Report

Comments and Suggestions for Authors

In this paper, the important parameters of particle concentration in characterizing gas-solid two-phase flow are explored by comparing and analyzing three methods, namely, BP neural network, Recurrent Neural Network, and Long Short-Term Memory. Seven parameters including temperature, humidity, upstream and downstream sensor signals, delay, pressure difference and particle concentration were used as the data set for the experiments. Evaluation metrics, such as prediction accuracy, were used to assess the performance of the methods. I recommended this work can be published. This paper can be accepted, however, there are some points, which need to be clearly revised and explained, shown as below:

1.      The subscripts and superscripts in the formulas should be clear and distinguishable, and their sizes should be appropriate. For example, in equations (1) and (2), please make the necessary modifications. Additionally, please review and ensure that similar errors are not present throughout the entire revised manuscript.

2.      In line 105 of the manuscript, the citation format for reference [6] differs from that of other references. Please compare it with the paper's formatting requirements and ensure consistency in the citation format for references throughout the entire manuscript.

3.      In section “2.2.2. LSTM” of the manuscript, ensure consistency in the formatting of the following sub-sections: 1. forget gate, 2. input gate, 3. Updating Memory State, and 4. Updating Memory State.

4.      The full form of abbreviations should be written out starting from the abstract, and thereafter, abbreviations can be used in the manuscript. In section “4.1 Model Evaluation Metrics”, there is a repetition of the abbreviation for “Root Mean Square Error (ERMSE) and Mean Absolute Error (EMAE)” at line 195 and 209. Please review the entire manuscript and correct such errors.

5.      Please revise the “1. Introduction” section of the manuscript, as it currently references only two citations and lacks sufficient depth. Please refer to existing research literature and supplement this section with more comprehensive content.

6.      The mention of using seven parameters, including temperature, humidity, upstream and downstream sensor signals, delay, pressure difference, and particle concentration as the experimental dataset is quite brief in the manuscript. It would be beneficial to provide additional details on the experimental design, equipment employed, types of sensors used, and the size of solid particles in the experiment. Please elaborate on the experimental process to enhance the experimental content of the manuscript.

7.      In section “4.3 Results”, the prediction results of the BP neural network (BPNN), Recurrent Neural Network (RNN), and Long Short-Term Memory (LSTM) models for the experimental dataset are 92.5, 92.4, and 92.7, respectively. Considering the small differences in prediction accuracy, if it is argued that the LSTM model exhibits better performance, please provide an explanation.

Comments on the Quality of English Language

Moderate editing of English language required

Author Response

1. Summary  

Thank you for taking the time to review our manuscript titled [Application of Deep learning in Predicting Particle Concentration of Gas-Solid Two-Phase Flow]. We would like to express our sincere gratitude for your thorough evaluation and insightful comments. Your feedback has been invaluable in improving the quality of our work.

We have carefully considered each of your comments and have made the necessary revisions as per your suggestions. The detailed responses to your comments and the corresponding revisions highlighted in the track changes have been included in the re-submitted files.

We greatly appreciate your constructive feedback, which has undoubtedly strengthened the rigor and clarity of our manuscript. We are confident that these revisions have enhanced the overall quality of the manuscript, and we hope that our efforts meet with your approval.

We look forward to the opportunity to address any further concerns and to the possibility of sharing our improved work with the academic community.

2. Questions for General Evaluation

Reviewer’s Evaluation

Response and Revisions

Does the introduction provide sufficient background and include all relevant references?

Can be improved

The introduction has been refined to provide comprehensive background information

Are all the cited references relevant to the research?

Yes

Is the research design appropriate?

Yes

Are the methods adequately described?

Yes

Are the results clearly presented?

Can be improved

We have revised the presentation of the research results and added additional explanatory notes.

Are the conclusions supported by the results?

Yes

3. Point-by-point response to Comments and Suggestions for Authors

Comments 1: [Please revise the “1. Introduction” section of the manuscript, as it currently references only two citations and lacks sufficient depth. Please refer to existing research literature and supplement this section with more comprehensive content.]

Response 1: [Thank you for pointing that out. I agree with this comment. Therefore, I made significant changes to the introduction, summarizing existing research.]

Comments 2: [The mention of using seven parameters, including temperature, humidity, upstream and downstream sensor signals, delay, pressure difference, and particle concentration as the experimental data set is quite brief in the manuscript. It would be beneficial to provide additional details on the experimental design, equipment employed, types of sensors used, and the size of solid particles in the experiment. Please elaborate on the experimental process to enhance the experimental content of the manuscript.]

Response 2: Agree. As a result, we have added details about the experimental procedure, including information on the experimental design, the equipment used, and the types of sensors employed. In the revised manuscript, this change can be found in lines 197-255.

Comments 3: [In section “4.3 Results”, the prediction results of the BP neural network (BPNN), Recurrent Neural Network (RNN), and Long Short-Term Memory (LSTM) models for the experimental data set are 92.5, 92.4, and 92.7, respectively. Considering the small differences in prediction accuracy, if it is argued that the LSTM model exhibits better performance, please provide an explanation.]

Response 3:The LSTM model exhibits superior memory and long-term dependency capabilities compared to the BPNN and RNN, making it suitable for handling time series data and long-distance dependency relationships, thereby providing the LSTM model with stronger modeling capabilities for gas-solid two-phase flow data processing.In the revised manuscript, this change is explained in the concluding section of the “4.4 Results” section, specifically at line 378.

4. Response to Comments on the Quality of English Language

Point 1:Moderate editing of English language required

Response 1:  I have made some edits to improve the expression in the article to ensure it meets the required standards.  

5. Additional clarifications

[I have made the necessary modifications regarding the other issues you mentioned about the paper format. Thank you very much for your patience and guidance]

Reviewer 2 Report

Comments and Suggestions for Authors

This paper focused on comparing three deep learning methods—BP neural network, RNN, and LSTM—for predicting particle concentration in gas-solid two-phase flows. It used a dataset comprising parameters like temperature, humidity, sensor signals, delay, pressure difference, and particle concentration. The study found similar prediction accuracies for all three methods, suggesting further research for optimization.

1. In Introduction, the authors stated that conventional data processing approaches face challenges in managing complex and nonlinear interconnections, exhibit high computational demands, and often lack precision in forecasting outcomes for multivariable data in gas-solid two-phase flow scenarios. To better emphasize the importance of current study, it suggests providing a more in-depth review of previous methods and their limitations with proper references cited.

2. Also, it would benefit the readers of this article, if the authors could discuss/review the current applications of deep learning in particle concentration prediction.

3. Based on my knowledge, backpropagation is a gradient estimation method used to train neural network models. Can the authors provide reference regarding the BP neural networks?

4. For Materials and Methods section, please provide references for all three neural network models used in this work as they were already well developed. Please clearly notify the original contribution from current work to the neural network models.

5. Line 105, extra period. Line 205, typo?

6. Only first 6 references were cited in the main text. Please properly cite the rest of the references in the main article. Also, I notice different citation formats exist in the text (line 105 and 110). Please fix.

7. How were the parameters in Table 1 determined? What method was used?

8. Based on Eq. (18), prediction accuracy A_F should be a serial values. Why there is only one single value of A_F in Table 2?

9. What is relative error interval (Figure 4)?

Comments on the Quality of English Language

Overall the English written is fine but a lot typos are noted in the current manuscript. Please proof read the entire article discretely. Also, please avoid first-person perspective, such as using the word "I".  

Author Response

Response to Reviewer Comments

1. Summary

Thank you for taking the time to review our manuscript titled [Application of Deep learning in Predicting Particle Concentration of Gas-Solid Two-Phase Flow]. We would like to express our sincere gratitude for your thorough evaluation and insightful comments. Your feedback has been invaluable in improving the quality of our work.

We have carefully considered each of your comments and have made the necessary revisions as per your suggestions. The detailed responses to your comments and the corresponding revisions highlighted in the track changes have been included in the re-submitted files.

We greatly appreciate your constructive feedback, which has undoubtedly strengthened the rigor and clarity of our manuscript. We are confident that these revisions have enhanced the overall quality of the manuscript, and we hope that our efforts meet with your approval.

We look forward to the opportunity to address any further concerns and to the possibility of sharing our improved work with the academic community.

2. Questions for General Evaluation

Reviewer’s Evaluation

Response and Revisions

Does the introduction provide sufficient background and include all relevant references?

Must be improved

The introduction has been refined to provide comprehensive background information

Are all the cited references relevant to the research?

Must be improved

We have carefully reviewed and made revisions to the referenced literature to ensure its relevance to our research.

Is the research design appropriate?

Yes

Are the methods adequately described?

Can be improved

We have supplemented the content of the experiment and provided a detailed description of the experimental steps.

Are the results clearly presented?

Yes

Are the conclusions supported by the results?

Yes

3. Point-by-point response to Comments and Suggestions for Authors

Comments 1: [In Introduction, the authors stated that conventional data processing approaches face challenges in managing complex and nonlinear interconnections, exhibit high computational demands, and often lack precision in forecasting outcomes for multivariable data in gas-solid two-phase flow scenarios. To better emphasize the importance of current study, it suggests providing a more in-depth review of previous methods and their limitations with proper references cited.Also, it would benefit the readers of this article, if the authors could discuss/review the current applications of deep learning in particle concentration prediction.]

Response 1: [Thank you for pointing that out. I agree with this comment. Therefore, I made significant changes to the introduction, summarizing existing research. ]

Comments 2: [How were the parameters in Table 1 determined? What method was used]

Response 2: The parameters in Table 1 were determined using an exhaustive search method. Detailed information can be found in the revised manuscript, specifically in the section titled “4.3 Model Construction and Parameter Determination” (lines 314-327).

Comments 3: Based on Eq. (18), prediction accuracy A_F should be a serial values. Why there is only one single value of A_F in Table 2?

Response 3 :As the assessment aimed to evaluate the overall performance of the model, the experiment employed average prediction accuracy. However, there were some issues with the formula expression, for which I apologize. The modifications have been made in the revised manuscript, specifically in the section titled “4.1 Model evaluation metrics.”

Comments 4:What is relative error interval (Figure 4)?

Response 4:I apologize for not providing a corresponding explanation for this part.After obtaining the predicted values and actual values of different models' test sets, the results can be divided into different intervals based on the relative error. By evaluating the proportion of low error interval samples, the performance of the models can be quantitatively assessed. A high proportion of low error interval samples indicates that the model can accurately predict the majority of samples, demonstrating a high level of precision.,And an explanation for this part has been added to line 362 of the revised manuscript.

4. Response to Comments on the Quality of English Language

Point 1:Overall the English written is fine but a lot typos are noted in the current manuscript. Please proof read the entire article discretely. Also, please avoid first-person perspective, such as using the word "I". 

Response 1: I have made some edits to improve the expression in the article to ensure it meets the required standards.

5. Additional clarifications

I have made the necessary modifications regarding the other issues you mentioned about the paper format. Thank you very much for your patience and guidance.

Round 2

Reviewer 2 Report

Comments and Suggestions for Authors

I appreciate that the authors revised the manuscript and addressed issues existed in previous version. However, some issues are not fully resolved in the updated manuscript. 

1. The written English improvement and discreet proofreading are required before the manuscript can be published. Please pay attention to the tense used in the text and the plural nouns used, for example, Figures 4 to 6 instead of Figure 4 to 6.

2. Issue with reference citation has not been properly addressed. Please cite the references following the guidelines for authors: https://www.mdpi.com/journal/fluids/instructions#editorial_procedure

Also, only the first 14 references are denoted in the main text. 

3. Is there a specific reason that 3 hidden layers were used for all three different models? Why was 500 used as the maximum number of iteration?

Comments on the Quality of English Language

See comments above.

Author Response

 Summary

Thank you sincerely for your valuable time and effort dedicated to reviewing our manuscript. We highly appreciate your constructive comments and suggestions. Below, you will find the detailed responses addressing each of the raised points, along with the corresponding revisions and corrections, which have been highlighted or tracked changes in the re-submitted files. We kindly request you to consider our incorporated changes and revisions, and we remain open to any further guidance or feedback you may provide. Once again, we express our gratitude for your contribution to improving the quality of our work.

Point-by-point response to Comments and Suggestions for Authors

Comments 1:The written English improvement and discreet proofreading are required before the manuscript can be published. Please pay attention to the tense used in the text and the plural nouns used, for example, Figures 4 to 6 instead of Figure 4 to 6.

Response 1:We will ensure the manuscript’s written English and plural nouns (e.g., Figures 4 to 6) are improved before publication. Thank you for your guidance.

Comments 2: Issue with reference citation has not been properly addressed. Please cite the references following the guidelines for authors:

Response 2: We will ensure proper adherence to the reference citation guidelines for authors. Thank you for bringing this to our attention.

Comments 3:Is there a specific reason that 3 hidden layers were used for all three different models? Why was 500 used as the maximum number of iteration?

Response 3:

The decision to use 3 hidden layers for all three models was based on experimental data and preliminary results, which demonstrated that models with three hidden layers possessed superior capability in capturing complex features within the data set. Conversely, we carefully considered the potential drawbacks of adding a fourth hidden layer, such as the heightened risk of over fitting and increased model complexity without significant performance improvement. As for the decision to cap the maximum iteration count at 500, this was informed by experimental data suggesting that the error of different models converged to a stable level after 500 iterations.This point can be reflected in lines 326-332 and lines 353-355 of the manuscript.